# Computer-based intervention for residents of domestic violence shelters with substance use: A randomized pilot study

Maji Hailemariam[1,2]*, Jennifer E. Johnson[1,2], Dawn M. Johnson[3], Alla Sikorskii[4], Caron Zlotnick[5,6,7]

**1** Charles Stewart Mott Department of Public Health, Michigan State University, College of Human Medicine, Flint, Michigan, United States of America, **2** Department of Obstetrics Gynecology and Reproductive Biology, Michigan State University, College of Human Medicine, East Lansing, Michigan, United States of America, **3** Department of Psychology, University of Akron, Akron, Ohio, United States of America, **4** Department of Psychiatry, Michigan State University, College of Osteopathic Medicine, East Lansing, Michigan, United States of America, **5** Department of Psychiatry and Human Behavior, Brown University, Providence, Rhode Island, United States of America, **6** Department of Medicine at Women and Infants Hospital, Providence, Rhode Island, United States of America, **7** Department of Psychiatry and Mental Health, University of Cape Town, Cape Town, South Africa

\* debenama@msu.edu

**Data Availability Statement:** All relevant data are within the manuscript and its Supporting information files.

## Abstract

### Background

Intimate Partner Violence (IPV) is a significant public health problem often associated with serious mental health and physical health implications. Substance use disorders (SUDs) are one of the most common comorbidities among women with IPV, increasing risk of subsequent IPV.

### Methods

The current study examined the feasibility, acceptability, and preliminary effectiveness of a brief computerized intervention to reduce alcohol and drug use among women with IPV. Fifty women with recent IPV and alcohol and drug use risk were recruited from domestic violence shelters and randomized to the experimental computerized intervention or to an attention and time control condition. The primary outcome was percent heavy drinking or drug using days in 3 month increments over the 6 months after leaving the shelter. Receipt of substance use services and IPV severity were evaluated as secondary outcomes.

### Results

The computerized intervention was feasible and acceptable, with high (n = 20, 80%) completion rates, engagement with the intervention, and satisfaction scores. As expected in this pilot trial, there were no significant differences between conditions in percent heavy drinking/drug using days or receipt of substance use services and large individual differences in outcomes. For example, receipt of substance use services decreased by a mean of 0.05 times/day from the baseline to the 6-month time period in the control condition (range -1.00

**Funding:** This study was funded by the National Institute of Drug Abuse (NIDA) grant number R34DA038770. The views expressed in this manuscript are those of the authors.

**Competing interests:** The authors have declared that no competing interests exist.

to +0.55) and increased by a mean of 0.06 times/day in the intervention condition (range -0.13 to +0.89). There were large decreases in IPV severity over time in both conditions, but directions of differences favored the control condition for IPV severity.

## Conclusion

A computerized intervention to reduce the risk of alcohol/drug use and subsequent IPV is feasible and acceptable among residents of a domestic violence shelter. A fully powered trial is needed to conclusively evaluate outcomes.

## Background

Intimate partner violence (IPV) is a pervasive and significant public health problem. It is often defined as violence by an intimate partner that may involve physical altercations (e.g., such as hitting, slapping or kicking), emotional or physical threats, and/or forced sexual relations [1]. Over 48 million women experience IPV in their lifetime, and about 41% of female IPV survivors report some form of physical injury due to the IPV [2]; including traumatic brain injury [3, 4]. The degree of morbidity associated with IPV is reflected in the fact that it negatively affects eight of the 10 leading health indicators identified by the Department of Health and Human Services (DHHS) [5].

Substance use problems are highly prevalent in women with IPV. Lifetime prevalence rates for substance abuse or dependence are twice as high for women with IPV than women in the general population [6, 7] and women with IPV are 5.6 times more likely to develop a substance use disorder compared to women not exposed to IPV. For women in U.S. domestic violence shelters, it has been estimated that 42% use substances [8, 9]. Moreover, given the multitude of competing concerns and high degree of stress faced by women with IPV in shelters and when they leave shelters, women with prior substance use problems appear to be at high risk of relapse. Common theories to explain the strong association between these two phenomena have included the self-medication hypothesis, that is, IPV survivors use substances to cope with the effects of violence [10–12]. Conversely, substance use can impair a woman's judgement or compromise her ability to move to safety [13, 14]. A third possibility is that substance use may increase conflict in relationships, leading to violent behavior towards the woman, especially if her partner is also using alcohol or drugs [15]. The pathways linking IPV and substance use are complex. In general, research supports a strong bidirectional relationship between these two morbidities [16, 17].

Comorbid substance use increases risk of new episodes of violence for women with IPV [18]. Negative consequences of IPV such as substance use can, in turn, impede a woman's ability to curtail future violence [19, 20]. Consistently, longitudinal studies have found that substance use is associated with new episodes of abuse in women with IPV [10, 21]. Furthermore, drug use is related to increased severity of physical partner victimization [16, 22, 23]. Given that cessation of violence is necessary for recovery from its traumatic effects, it seems imperative to address victimized women's substance use difficulties. Such an intervention may improve women's ability to effectively use resources to help break the cycle of violence and establish their and their children's safety.

There is consensus among experts that a critical time to provide substance use interventions for women with IPV is when they are seeking shelter services, because at these times women are open to changes in their lives and identities [24] and most likely to recover from violence

and from substance use [24]. A huge challenge for domestic violence shelters is that at current funding levels they cannot meet the overwhelming demand for services.

Digital interventions including mHealth and eHealth, may hold a strong promise of reducing IPV victimization and may improve mental health symptoms [25]. Therefore, they are generally considered feasible and acceptable [26]. For example, computer-delivered interventions can be delivered without substantial investments of time or energy from staff. Other advantages of computer-based interventions are that they are potentially low-cost and scalable while maintaining fidelity across sites. Therefore, computer-delivered interventions for residents of domestic violence shelters with substance use problems can overcome existing obstacles to addressing substance use in domestic violence shelters, particularly time, training, and cost challenges.

The current study presents findings from a pilot randomized controlled trial (RCT) of a brief computerized intervention that addresses known barriers in early identification and intervention for residents of domestic violence shelters with substance use issues. The current study examines the feasibility and acceptability (*primary*) of this brief a computerized program and presents preliminary evidence of whether the computerized intervention leads to improvements in substance use (heavy drinking or drug using), increased utilization of substance use services (both treatment and self-help utilization), and reduction in IPV severity over a 6- month post-shelter period compared to an attention-time matched control condition.

## Methods

### Participants, eligibility criteria, and settings

Participants were recruited from shelters in Rhode Island (n = 21), Ohio (n = 22) and Massachusetts (n = 7). Shelter staff informed research staff as new residents were admitted to the shelter. The research staff attended house meetings and scheduled times to be at the shelters when new residents were available and provided a description of the study. Shelter residents were recruited to participate in a computer-based survey to help women with IPV be healthy. Women who expressed interest were given the study information sheet. Women who showed interest in the study were screened for eligibility using the Computerized Intervention Authoring Software (CIAS) software delivered on an easy to use, ultraportable Tablet PC in a 10-15-minute screener or a computer in a private setting. The screener included a brief series of questions about general health, exercise, and diet. All screened participants received a standard health information brochure with further information and resources on the health topics mentioned, including a list of local substance use treatment referrals and community resources. Those meeting full inclusion criteria were asked to provide signed informed consent and completed the computer-based baseline assessments. Fig 1 presents the study flow chart.

Participants were included if they were 18 or older, residents of a domestic violence shelter, who were at risk substance users within the 3 months prior to entering the shelter, as determined by the screener, the NIDA-Modified ASSIST [27] and endorse IPV within the 3 months prior to entering the shelter, as determined by the screener, the Woman Abuse Screening Tool (WAST) [28]. Participants were excluded if they were unable to provide informed consent (e.g., due to florid psychosis or other clear cognitive impairment), 2) who did not understand English 3) who did not endorse on the baseline Timeline Follow Back (TLFB) for substance use and/or intimate partner violence in the 3 months prior to shelter stay. Women who had a prescribed medical marijuana card were also excluded from the study (if medical use was the only use). All women received standard shelter services, which primarily included case management and crisis intervention.

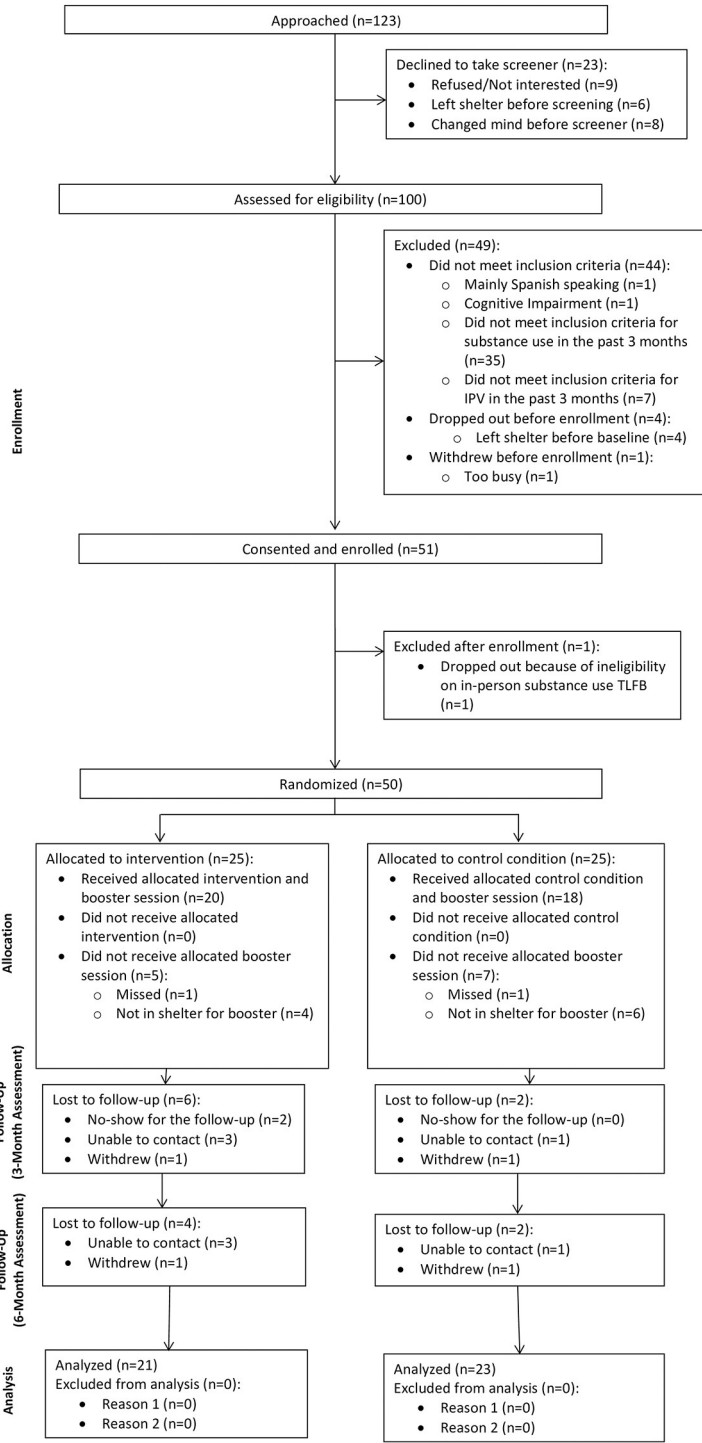

**Consort Table**

*Computer-Based Intervention for Sheltered Batter Women with Substance Use*
(Randomized Trial)

**Fig 1. The CONSORT diagram.**

## Interventions

**The computerized intervention.** The intervention was a computer-based, brief intervention to reduce substance use among residents of domestic violence shelters with IPV. The intervention used a sophisticated intervention development tool, the CIAS which is highly interactive with immediate responses to most input, occasional summaries, branching based on participant characteristics, responses, or preferences, and empathic reflections. The intervention consisted of a 50-minute intervention on the Tablet PC and a 15-minute "booster" session on the Tablet PC two weeks after the 50-minute intervention session to reinforce the effects of the intervention. Both sessions took place in a private shelter setting while the participant was a resident of the shelter. The computer-based intervention and booster session were conducted in a private space at the domestic violence shelter during participants' shelter stay. The intervention took place during participants' shelter stay because women are more likely to initiate changes when they are in a supportive environment. In addition, digital interventions raise privacy and safety concerns for women who live with their abusive partner.

Motivational strategies such as reflective listening and tailored summaries and feedback were utilized to enhance participants' motivation. The intervention presented personalized feedback from the baseline assessment. Next the intervention delivered psychoeducation about participant's substance use and associated risks for the woman, including partner substance use; the bidirectional relationship between substance use and IPV; and risks of untreated substance use such as increased risk of IPV. The intervention was tailored based on the current substance use status of each participant (see Fig 2). The intervention contained language referring to either substance use in general (past or present behaviors and beliefs) or current substance use and does not assume current substance use. For women who stated that they have already quit (see "No" in Fig 2), the focus was on how they can remain abstinent now and after they leave the shelter (Relapse Prevention). Women who endorsed current substance use were asked about their interest/readiness in quitting their substance use ("Ready to Quit?") leading to a bifurcated treatment response such that those participants reporting a goal of immediate abstinence were moved more quickly to a section consistent with primary goal-setting for substance use reduction. This arm included components of a quit contract and assist the woman in identifying specific goals and a timeline for reaching these goals. These participants had the option to create a personalized Safety Plan on paper that included the specific goals that the women have identified and/or optionally, can enter their own change goals in free text. Some examples of goals include attending AA or NA, attending substance use treatment, speaking to shelter staff about coordinating substance use treatment, planning for support groups after leaving the shelter such as IPV support groups. For the remaining participants who reported not being ready to quit at this time, the narrator presented sections including pros and cons, feedback, and optional goal setting. The interventions ended with a motivational video.

Within two weeks of the intervention, women completed a 15-minute computerized booster session in which they review the relevant components of the intervention session for each woman (e.g., pros and cons, feedback, and goal setting) and their own personalized safety plan. The booster session aimed to bolster the effects of the intervention. Women were given the option to revise their personalized Safety Plan and were allowed to select from a menu of potential personal change goals.

**Control condition.** Participants in the control condition also completed the same assessments. After randomization, participants in the control condition received a 50-minute session and a 15-minute booster session within two weeks after the initial session on the Tablet PC, which comprised of the viewing of popular entertainers/shows videos. This condition

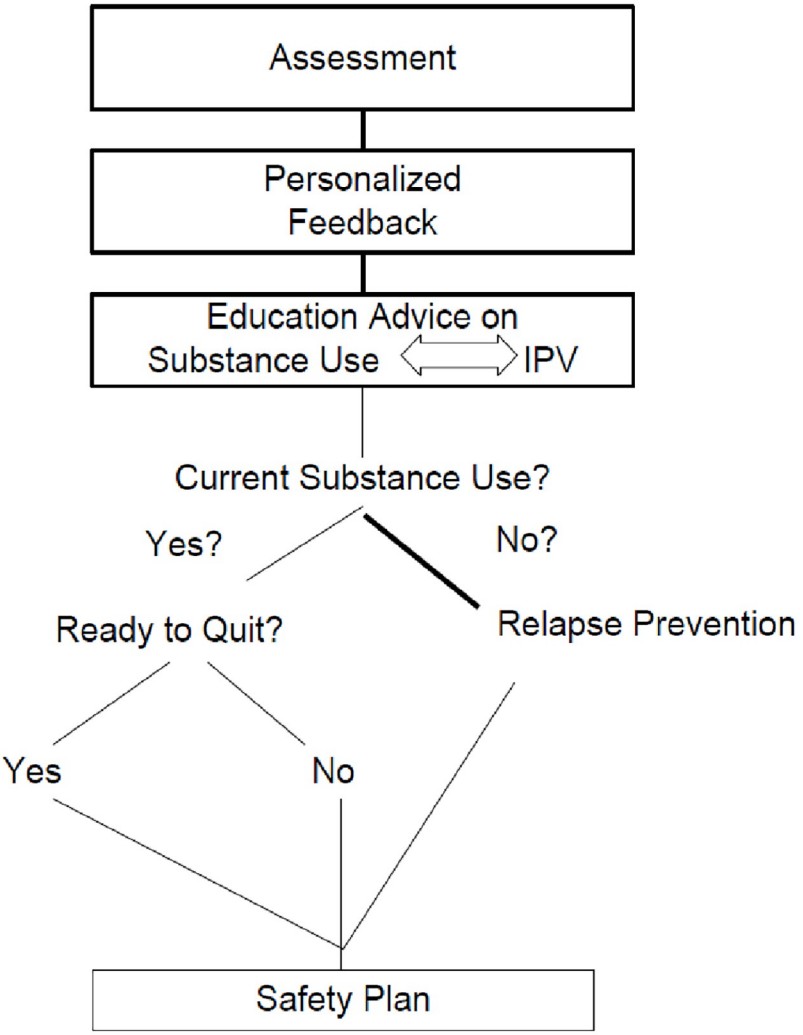

**Fig 2. The intervention process.**

controlled for time spent on the computer-based intervention, maintain blinding of research assistants, and mimic the interactivity of the computer-based intervention condition. The control condition was conducted in a private space at the shelter during participants' shelter stay. Throughout the study, participants received other standard services they are eligible for regardless of their intervention condition.

## Sample size and power analysis

For this intervention development pilot project, assessment of feasibility and acceptability of the intervention and research procedures was the primary goal. Nonetheless, pilot data can be used to demonstrate whether the effects of treatment look promising across a set of outcome variables, and to suggest, in concert with results from larger scale clinical trials in related fields, the range of effect sizes that would be reasonable to expect in a future trial. As a result, the goal was to obtain estimates of the differences between treatment conditions with 95% confidence intervals and effect size estimates. The actual enrollment was 50 participants with an 88% retention rate (for an evaluable sample of n = 44).

Given the sample size, the unadjusted effect size Cohen's d = 0.86 was detectable in between-group comparisons with power of .80 at .05 level of significance in two-sided tests. In longitudinal analysis of two repeated measures adjusted for baseline, with correlations between pairs of repeated measures ranging from 0.1 to 0.6, the detectable adjusted effect sizes ranged from 0.57 to 0.66. If the observed effect size was smaller, statistical significance in between-group comparisons would not be reached, however statistical significance was not the goal of this pilot study. Sample size guidelines for treatment development projects [29] such as this one recommend 15 to 30 participants per cell to produce reasonable estimates of the effect sizes. The available sample sizes of 21 and 23 in each condition are within this recommendation. No interim analyses or stopping guidelines were planned in this small pilot trial.

**Randomization (random number generation, allocation concealment, implementation).** The computer program randomized each participant on 1:1 basis in real time as participants entered the study. After completion of the baseline assessment, the (computer) narrator "flipped a coin" and women (N = 50) were randomized into the control or intervention condition.

## Blinding

Because randomization and both intervention conditions were computer delivered, research staff were blind to intervention condition throughout the study.

## Assessments

The study presents preliminary evidence of whether the computerized intervention leads to improvements in substance use (heavy drinking or drug using), increased utilization of substance use services (both treatment and self-help utilization), and reduction in IPV severity over a 6- month post-shelter period compared to the control condition.

**Feasibility and acceptability.** We assessed feasibility of the research procedures by examining study recruitment and refusal rates, participants' willingness to be randomized, and follow-up rates. We assessed the feasibility and acceptability of the Computerized intervention by examining rates of intervention completion. Our goal was that 80% of participants would complete both baseline and booster sessions. We also used the *Satisfaction with CIAS Software Scale* [30] (i.e., an average score of 4 out of 5 on the CIAS satisfaction scale; at least highly satisfied and engaged with software) and *Client Satisfaction Questionnaire-Revised (CSQ-8-R)* [31] (i.e., above the average score of 27 on the CSQ-8-R) to assess acceptability of the Computerized intervention. *Satisfaction with CIAS Software Scale (SSS)* assesses participant satisfaction on items tapping on likeability, ease of use, level of interest, and respectfulness [30]. The well-studied 8-item *CSQ-8-R* assesses intervention satisfaction [31]. These scales were administered after each computer-based session (i.e., initial session and booster session).

**Demographics and screening.** All participants completed demographic information, including age, race, ethnicity, educational level, marital status and employment (status, # hours per week). *The NIDA-Modified Alcohol, Smoking, Substance Involvement Screening Test (NIDA-Modified ASSIST* [27] was used to assess for alcohol use, illegal drug use, and nonmedical prescription drug use in the past three months prior to shelter stay. After eligibility was confirmed, those who were considered at risk were categorized as an at-risk drinker and/or at moderate or high risk for substance use based on the NIDA-Modified ASSIST [27] criteria for these risk categories. At screening, IPV in the past three months prior to shelter stay was assessed with the WAST [28]. The WAST is an 8-item instrument that measures physical, sexual, and emotional abuse and is consistent with the definition of IPV as defined by The American College of Obstetricians and Gynecologists [28, 32]. It has correctly classified 100% of

non-abused women and 92% of abused women in a known-group analysis [28], has good internal reliability [33], and has adequate concurrent validity, even with ethnic minorities [33]. A cut off point of $\geq 4$ was used to determine the presence of IPV.

## Outcomes

**Primary outcome.** Our primary outcome was the percent of heavy drinking or drug using days in 3 month increments over the 6 months after leaving the shelter, assessed using the Timeline Follow-Back (TLFB)-modified computer version [34, 35], administered in person. Baseline TLFB assessment covered the 90 days prior to entering the shelter. For primary analysis, days using drugs and heavy drinking days were combined to create a single variable that reflected the total number of days that woman used drugs or had 4+ drinks. The TLFB has excellent reliability [36], and is sensitive to change as used other studies [37, 38]. Breath alcohol tests were conducted at each follow-up assessment (i.e., at the 3 months and 6 months periods) to ensure that the client had consumed no alcohol prior to the assessment (breath alcohol content < .02). Assessments were rescheduled if the participant had been consuming alcohol.

**Secondary outcomes.** Secondary outcomes included use of substance use services and IPV severity over a 6-month post-shelter period. The Treatment Services Review (TSR) [39] was used to assess substance use services (both treatment and self-help utilization) received (including outpatient, day patient, residential treatment, NA, AA) to capture the extent to which women reached out to access recovery-related resources. At the 3 time points, the TSR was used to assess substance use services used in the 90 days prior to entering the shelter, since leaving the shelter, and since the last assessment, respectively. Because total days during each follow-up time point varied from participant to participant, days in the follow-up period was used as an offset in analyses predicting number of substance use treatment sessions attended.

Severity of IPV was assessed using several measures assessed for the 90 days prior to baseline and at each follow-up time point. The primary IPV severity measure was the Composite Abuse Scale (CAS). The CAS is a widely used self-report of behaviors scale with 4 subscales that measure severe, combined abuse, emotional abuse, physical abuse, and harassment. The CAS has been published in the Centers for Disease Control and Prevention compendium of intimate partner violence measures [40]. It consists of 30 items presented in a six-point format requiring respondents to answer "never", "only once", "several times", "monthly", "weekly" or "daily" in a twelve-month period [40]. It includes a victimization scale and a perpetrator scale. A woman's adoption of safety behavior was evaluated using Safety Behavior Checklist (SBC) [41, 42]. SBC has 15 items that assess the use of strategies suggested to keep victim safe (e.g., hiding money and extra clothing). To measure another aspect of IPV, cyberstalking, a 6-item scale based on the prevailing literature on cyberstalking was administered. This cyberstalking measure assessed cyberstalking behaviors and was presented in the same 6-point format as the CAS.

**Theorized mediators of intervention effects on substance use outcomes.** The Confidence Ladder [43–45] was used to assess the participants' levels of confidence regarding their ability (self-efficacy) to change and abstain from substances. Readiness Ruler [46, 47] was used to assess the degree to which participants' are ready to cut down or quit alcohol or drugs.

## Statistical analyses

Feasibility and acceptability measures were examined descriptively. Unadjusted comparisons of outcomes at 3 and 6 months were performed using t-tests or non-parametric tests (i.e., the Wilcoxon rank-sum tests for independent samples) for outcomes not approximately normally distributed, such as count-like variables (i.e., percent heavy drinking or drug using days, substance use services received per day). Adjusted analyses used longitudinal modeling for

repeated measures of the outcomes (3 and 6 months) with study group as the independent variable and baseline value as a covariate. The technique for longitudinal modeling was generalized estimating equations with weighted estimation approach that allowed for data missing at random, so that all participants with at least one non-missing follow-up observation were included. The error distribution was chosen based on each outcome's distribution: Normal (scale score; readiness and self-efficacy), or Negative Binomial for over-dispersed count outcomes (all other outcomes). In the Negative Binomial models, log link was used, and differences between logarithms of group means were evaluated. To facilitate the interpretation on the original measurement scale, the estimated of logarithms were then back transformed (exponentiated) to the original scale. As a result of exponentiation, the differences between logarithms of two group means corresponded to the ratios of exponentiated estimates of the means. We obtained 95% confidence intervals for the differences of group means or logarithms of the means along with effect sizes (Cohen's d) for the differences between two groups.

### Consent and trial registration

The Intervention for Battered Sheltered Women with Substance Use Randomized Trial study was approved by Women and Infants Hospital of Providence, Rhode Island. All study staff completed human subject research and good clinical practice training modules. The study staff obtained written informed consents. The study was also registered in www.clinicaltrials.gov under identifier # NCT02629133, date of registration 09 December 2015, https://clinicaltrials.gov/ct2/show/NCT02629133.

## Results

A total of 50 participants were enrolled in the study. The racial profile of the participants included 42% (n = 21) Black, 38% (n = 19) White, 18% (n = 9) Latina and other 2% (n = 1). The rest identified as biracial, Native American or "other". Ages of participants ranged from 18 to 62 years. The median age of participants was 35.2 years. About 36% (n = 18) of the participants were never married and 42% (n = 21) were living together but not married. Sociodemographic details of participants are presented in Table 1.

### Feasibility and acceptability

**Feasibility of research procedures.** Study recruitment, refusal, and follow-up rates are shown in Fig 1.

**Feasibility and acceptability of the computerized intervention.** With 20 of the 25 participants successfully completing the intervention, the study achieved its planned 80% completion rate. Mean CSQ-8-R scores in the computerized condition were 29.9 (standard deviation [SD] = 0.62) on a scale of 8 to 32, exceeding the scale average score of 27. Mean CAIS satisfaction scale scores in the computerized intervention condition were 4.78 (SD = 0.28) indicating high satisfaction and engagement with the computerized intervention.

### Percent heavy drinking/drug using days (primary outcome)

There were no significant differences between intervention and control groups at 3 or 6 months in the cross-sectional unadjusted (S1 File) or adjusted longitudinal (Table 2) analyses. There were wide individual differences in outcomes. For example, from baseline to 3 months, percent drinking days decreased by a mean of 20% (range of decreased by 70% to increased 34%) in the control condition and decreased by a mean of 23% (range of decreased by 87% to increased by 66%) in the intervention condition.

**Table 1. Sociodemographic characteristics of participants.**

| Ethnicity | |
|---|---|
| Non-Hispanic/Latino | 41 (82%) |
| Hispanic/Latino | 9 (18%) |
| **Race** | |
| Black | 21 (42%) |
| White | 19 (38%) |
| Latina | 9 (18%) |
| Other | 1 (2%) |
| **Age** | |
| Range 18–62, average 35 | |
| **Education** | |
| Completed high school or has GED | 14 (28%) |
| Completed high school or above | 36 (72%) |
| **Marital status** | |
| Never been married | 18 (36%) |
| Married | 6 (12%) |
| Living together but not married | 21 (42%) |
| Separated or divorced | 5 (10%) |
| **Employment status** | |
| Currently employed | 9 (18%) |
| Unemployed | 41 (82%) |

**Table 2. Effects of experimental condition (average over time) on outcomes, adjusted for baseline value of each outcome.**

| | Control condition adjusted log mean (standard error) | Computerized intervention condition adjusted log mean (standard error) | Difference between logs of the means (95% confidence interval) | Control condition exponentiated adjusted log mean (SE) | Computerized intervention condition exponentiated adjusted log mean (SE) | Ratio of exponentiated adjusted log means, 95% CI | p | Effect size Cohen's d |
|---|---|---|---|---|---|---|---|---|
| Composite Abuse Scale–Victimization (lower is better) | 1.85 (0.35) | 2.75 (0.18) | -0.90 (-1.69, -0.10) | 6.37 (2.20) | 15.51 (2.89) | 0.41 (0.18, 0.90) | **.03** | 0.71 |
| Composite Abuse Scale–Perpetration (lower is better) | 0.46 (0.32) | 1.23 (0.26) | -0.77 (-1.59, 0.46) | 1.58 (0.50) | 3.32 (0.89) | 0.46 (0.20, 1.05) | .06 | 0.59 |
| Cyber-stalking Scale score (lower is better) | 0.94 (0.23) | 1.35 (0.25) | -0.41 (-1.12, 0.30) | 2.56 (0.59) | 3.86 (0.98) | 0.66 (0.33, 1.35) | .26 | 0.36 |
| Safety Behavior Checklist total score (higher is better) | 1.62 (0.18) | 1.29 (0.15) | 0.33 (-0.13, 0.80) | 5.06 (0.90) | 3.63 (0.56) | 1.39 (0.88, 2.22) | .16 | 0.45 |
| TLFB heavy drinking/drug using days (lower is better) | 3.26 (0.19) | 3.41 (0.27) | -0.15 (-0.79, 0.48) | 25.93 (4.93) | 30.16 (8.04) | 0.28 (0.46, 1.62) | .64 | 0.15 |
| TSR number of substance use treatment times (higher is better) | -3.54 (0.57) | -2.75 (0.43) | -0.79 (-2.17, 0.45) | 0.03 (0.02) | 0.06 (0.03) | 0.45 (0.12, 1.81) | .26 | 0.36 |

## Secondary outcomes

**Receipt of substance use services.** There were no significant differences between intervention and control groups at 3 or 6 months in the cross-sectional unadjusted (S1 File) or adjusted longitudinal (Table 2) analyses. Again, there were large individual differences in outcomes. Receipt of substance use services decreased by a mean of 0.05 times/day from the baseline to the 6-month time period in the control condition (range -1.00 to +0.55) and increased by a mean of 0.06 times/day in the intervention condition (range -0.13 to +0.89).

**IPV severity.** Controlling for baseline values, CAS Victimization scores were significantly worse and CAS–Perpetration scores trended toward being worse over follow-up in the intervention relative to the control condition (see Table 2). Differences seemed to be driven by differences at the 6 month follow-up (see S1 File). Controlling for baseline values, analysis of the 15-item Safety Behavior Checklist (with higher scores reflecting better adoption of safety behaviors) and of Cyber-Stalking Scale scores over follow-up showed no differences between conditions (see Table 2). However, CAS Victimization, CAS Perpetration, and Cyber-Stalking Scale scores improved dramatically over time in both conditions. For example, median baseline CAS Victimization scores were over 50 in both conditions at baseline, and less than 10 across both follow ups in both conditions.

## Theorized mediators

Controlling for baseline values, analysis of Readiness Ruler scores (reflecting readiness to cut down or quit alcohol/drugs) over follow-up showed no differences between conditions. Controlling for baseline values, analysis of the Self-Efficacy Ladder scores (indicating self-efficacy in reducing substance use) also showed no differences between conditions (see Table 3). Because the computerized intervention did not significantly increase either proposed mediator or the ultimate outcome (i.e., substance using days), no further mediation analyses were undertaken.

## Discussion

In this study, we evaluated the feasibility and acceptability of a brief computerized intervention for residents of domestic violence shelters with substance use disorders who also report IPV in the past three months. Results suggested that a computerized intervention to improve SUD is feasible and acceptable among women with IPV and substance use problems at domestic violence shelters across 3 U.S. states (RI, MA and OH).

Results support the feasibility and acceptability of the brief computerized intervention for women with SUD in domestic violence shelters. This was demonstrated by the high (80%) study completion rate and higher rates of client satisfaction with the intervention as measured

**Table 3. Effects of experimental condition (average over time) on theorized mediators, adjusted for baseline value of each outcome.**

| | Control condition adjusted mean (SE) | Computerized intervention adjusted mean (SE) | Difference between adjusted means, 95% CI | p | Effect size Cohen's d |
|---|---|---|---|---|---|
| Self-efficacy to reduce substance use (higher is better) | 4.37 (0.20) | 3.94 (0.20) | 0.43 (-0.12, 0.98) | .12 | 0.49 |
| Readiness to reduce substance use (higher is better) | 3.25 (0.38) | 3.72 (0.23) | -0.47 (-1.33, 0.40) | .29 | 0.34 |

Note: Highlighted squares indicate (non-significantly or significantly) more favorable values; Normal error distribution in the analysis of self-efficacy and readiness; NB distribution for the rest.

by the CSQ-8-R. The high CIAS scale satisfaction scores also indicate the overall satisfaction with the program and engagement with the computerized intervention.

Within the context of a small pilot trial which was designed to provide a preliminary test of the feasibility and acceptability of a new intervention, the main conclusions that can be drawn are about feasibility and acceptability. Any preliminary efficacy results should be interpreted with extreme caution. Individual variation for almost all outcomes was wide, and given small sample size, the resulting confidence intervals for the differences were very wide, nearly one baseline standard deviation in length (Table 3, S1 File). In the negative binomial modeling shown in Table 2, Cohen's d was estimated based on the differences between estimated logarithms of the means. When the estimates of the logarithms were exponentiated, the width of resulting confidence intervals for the ratios of the means of two conditions was one to 4 times the magnitude of the ratio itself. As expected, due to its pilot nature, this pilot trial did not produce statistically significant differences in percent heavy drinking/drug using days between conditions. Point estimates favored the intervention condition for more substance use treatment received and readiness to reduce substance use. The study was not designed to provide statistical power to formally compare conditions; a subsequent fully-powered trial is the needed to provide definitive efficacy conclusions.

In this pilot trial, we measured IPV severity in the domains of victimization, perpetration, cyberstalking and safety behavior, there were large decreases in IPV severity over time in both conditions. However, effect sizes tended to favor the control condition for IPV severity. This finding (where IPV variables declined overall regardless of the intervention status and in some cases trended in favor of the control condition) is consistent with what was reported by other brief safety interventions for women with history of IPV and SUD [48–50]. A modification of the intervention might be required in the future to enhance its impact on IPV variables. Given that all participants were in a shelter, participants in both groups had an overall lower risk of severe IPV. It is also possible that the dose received of the computerized intervention was much smaller than the other (shelter-based) services women were receiving.

## Strengths and limitations

This study followed a rigorous design, used validated measures and a robust/randomized approach to testing the feasibility and acceptability of a brief computerized intervention to reduce SUD in women with IPV. The goal of this study was to establish feasibility and acceptability of the computerized intervention. Therefore, it was not powered for effectiveness outcomes, and effectiveness results should be interpreted with caution [51]. Moreover, the use of self-reports, assessment reactivity and effect on measured outcomes, the inclusion of only English-speaking participants are also limitations of this study.

## Conclusion

Computerized intervention to improve SUD is feasible and acceptable among women with IPV with substance use disorders at domestic violence shelters across 3 states. Further effectiveness testing in a larger sample is needed.

## Supporting information

**S1 File. Summary of unadjusted outcomes at baseline, 3 and 6 months by study condition.**
(DOCX)

**S2 File. IRB protocol for Women and Infants Hospital.**
(DOCX)

**S1 Checklist. The CONSORT checklist.**
(DOC)

## Author Contributions

**Data curation:** Caron Zlotnick.

**Formal analysis:** Jennifer E. Johnson, Alla Sikorskii.

**Funding acquisition:** Caron Zlotnick.

**Investigation:** Caron Zlotnick.

**Methodology:** Maji Hailemariam, Jennifer E. Johnson.

**Writing – original draft:** Maji Hailemariam.

**Writing – review & editing:** Maji Hailemariam, Jennifer E. Johnson, Dawn M. Johnson.

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
