## [Decision Letter · Decision Letter 0]

1 Dec 2022

PONE-D-22-28122Computer-based intervention for residents of domestic violence shelters with substance use: A randomized pilot studyPLOS ONE

Dear Dr. Hailemariam,

Thank you for submitting your manuscript to PLOS ONE. After careful consideration, we feel that it has merit but does not fully meet PLOS ONE’s publication criteria as it currently stands. Therefore, we invite you to submit a revised version of the manuscript that addresses the points raised during the review process. Your manuscript has been assessed by two expert reviewers, whose comments are appended below. The reviewers have highlighted multiple areas in which additional information or discussion is required regarding the study design, statistical analysis, and the rationale for your study with respect to existing literature. Please ensure you respond to each of these points carefully in your response to reviewers document, and revise your manuscript accordingly.  

We look forward to receiving your revised manuscript.

Kind regards,

Dr Joseph Donlan

Senior Editor

PLOS ONE

Journal Requirements:

"This study was funded by the National Institute of Drug Abuse (NIDA) grant number R34DA038770. The views expressed in this manuscript are those of the authors."

4. We note that the original protocol that you have uploaded as a Supporting Information file contains an institutional logo. As this logo is likely copyrighted, we ask that you please remove it from this file and upload an updated version upon resubmission.

Reviewers' comments:

Reviewer's Responses to Questions

**Comments to the Author**

1. Is the manuscript technically sound, and do the data support the conclusions?

Reviewer #1: Partly

Reviewer #2: Partly

2. Has the statistical analysis been performed appropriately and rigorously? 

Reviewer #1: Yes

Reviewer #2: Yes

3. Have the authors made all data underlying the findings in their manuscript fully available?

Reviewer #1: Yes

Reviewer #2: Yes

4. Is the manuscript presented in an intelligible fashion and written in standard English?

Reviewer #1: Yes

Reviewer #2: Yes

5. Review Comments to the Author

Reviewer #1: This manuscript presents data analysis from a randomized pilot study on computer-based ntervention to reduce alcohol and drug use among women with IPV. The topic is of importance, the study was registered as a RCT with a valid NCT number, and was approved by the respective IRB/Ethics Committee. While the study objectives sound interesting, is important, and on target, some shortcomings were observed, in regards to abiding by the CONSORT guidelines for conducting and reporting results of high-quality randomized controlled trials (RCTs). Some other (statistical) comments were also provided.

1. Methods:

Methods reporting need some work. An orderly manner is suggested, following CONSORT guidelines, without repeating information, such as Trial Design, Participant Eligibility Crtieria and settings, Interventions, Outcomes, sample size/power considerations, Interim analysis and stopping rules, Randomization (details on random number generation, allocation concealment, implementation), Blinding issues, etc, should be mentioned. The authors are advised to create separate subsections for each of the possible topics (whichever necessary), and that way produce a very clear writeup. I see the Authors indeed made an attempt; however, they are advised to write it carefully, following nice examples in the manuscript below:

https://www.sciencedirect.com/science/article/pii/S0889540619300010

Specific comments:

(a) For instance, the randomization and allocation concealment should be made very clear (they are NOT the same thing); the trial staff recruiting patients should NOT have the randomization list. Randomization should be prepared by the trial statistician, and he/she would not participate in the recruiting.

(b) The manuscript does not provide details on the randomization procedure; it just states as a "standard" procedure. Provide more details. Why was block randomization (BR) used? If not, any specific reason, why? This is because BR is often recommended in clinical studies/trials to ensure a balance in sample size across groups over time.

https://www.ncbi.nlm.nih.gov/pmc/articles/PMC2267325/

(c) Sample size/power: A sample size/power statement is still required even in a randomized pilot trial; see

https://www.sciencedirect.com/science/article/abs/pii/S0895435612002740

It's better if that is provided, utilizing the primary response, acceptable effect size, and at 90%, or 95%. Even pilot trials need to be conducted with some ballpark number. It should also be described in a separate sub-section.

(d) Statistical Analysis: Based on the (longitudinal) design of the study, the authors justifiably conducted a mixed linear model analysis. Who were the underlying Gaussian assumptions (of the random effect & error) assessed? Furthermore, a generalized estimating equation (GEE) would also be adequate; any reason why that was not conducted? (I am not asking authors to do it).

2. Results & Conclusions:

(a) The authors should check that any statement of significance should be followed by a p-value in the entire Results section. Otherwise, the Results section look OK.

Reviewer #2: The study is addressing two important public health challenges in women's health, IPV and substance use. The use of technology based interventions in the domestic violence shelter is justified given the multiple needs and overburdened staff. However, the manuscript would be strengthened with more details on the intervention (screen shots), details on the content related to substance use, given there may be use of multiple substances, etc. For example, what skills building, etc. are the focus for behavioral change? Does the intervention provide the user with specific local referrals to service organizations in the community, what is the link between the intervention content and increase daily substance abuse treatment use? The intervention describes a personalized safety plan, does this mean that the safety plan includes strategies to seek treatment or discussing substance use with counselors for example? Are advocates/case managers in the shelter trained on the intervention, would be likely that a participant would discus the information provided for behavior change and/or discuss/review personalized safety plan with an advocate, how was the advocate involvement in supporting participant taking into consideration when examining the implications of the findings? More detail on the choice of control condition is needed, as it appears the control condition is not related to behavioral change but rather on time that equal intervention group. All the women are in shelter, it would be interested to understand what other services they are receiving during the stay that may also influence findings, such as case management, legal support, child care, employment assistance, substance abuse treatment referrals, etc. More discussion is needed related to increase in self-efficacy in control group. Further, more discussion is needed related to the findings on IPV, it appears that IPV perpetration over time has a pattern towards significance with intervention group reporting more perpetration than control group, and control condition appears to have greater reduction in IPV victimization, cyberbullying and perpetration than intervention, although not significant - it would be interested in more discussion about these findings given IPV severity is stated outcome. Further, would be useful to report that participants in control condition used more safety strategies than intervention, this may not matter as use does not indicate that strategy was helpful/useful, so think need more discussion of these findings and how they inform future versions of the intervention. For example, the lower use of strategies by intervention group may be related to the personalized strategies so intervention participants did not need to use more strategies.

I do think detailing a framework for feasibility/acceptability testing would be useful, what was the measure used for acceptability for the intervention? I would recommend a table of the feasibility/acceptability, outcome and mediator measures used for study, would make it easier to read and follow. More information on human subjects protections given that many shelters do not allow women to use substance, what strategies were used if a woman was active user?

The study did not detail limitations, small non generalizable sample from 3 different shelters, no race/ethnic difference assessed related to substance use, IPV severity, etc. No discussion of reasons why control group had significantly improved self-efficacy? Although the intervention was reviewed as feasible, there was little indication that the intervention would be effective given the outcomes of reduce substance use, IPV severity and self-efficacy, what additional work, like qualitative interviews, etc will need to be done to revise the intervention to meet the needs of survivors?

6. PLOS authors have the option to publish the peer review history of their article (what does this mean?). If published, this will include your full peer review and any attached files.

Reviewer #1: No

Reviewer #2: No

---

## [Author Response · Author response to Decision Letter 0]

10 Apr 2023

Date: 04/07/2023 

To the Editor in Chief

PLOS ONE 

Subject: Submission of Original Clinical Trial Manuscript for Publication 

Dear Editor,

On behalf of the coauthors and myself, I would like to submit our revised manuscript titled “Computer-based intervention for residents of domestic violence shelters with substance use: A randomized pilot study” to be considered for publication. We thank the reviewers and the editor for the extensive feedback that has strengthened our manuscript. 

We have thoroughly addressed the concerns raised by the editor and the reviewers. Below are the point-by-point responses to the concerns along with a specific reference to changes made within the manuscript. In this revision, we have also invited in a co-author who is statistician who assisted in the reanalysis of the data. 

5. Review Comments to the Author

Reviewer #1: This manuscript presents data analysis from a randomized pilot study on computer-based intervention to reduce alcohol and drug use among women with IPV. The topic is of importance, the study was registered as a RCT with a valid NCT number, and was approved by the respective IRB/Ethics Committee. While the study objectives sound interesting, is important, and on target, some shortcomings were observed, in regards to abiding by the CONSORT guidelines for conducting and reporting results of high-quality randomized controlled trials (RCTs). Some other (statistical) comments were also provided.

1. Methods:

Methods reporting need some work. An orderly manner is suggested, following CONSORT guidelines, without repeating information, such as Trial Design, Participant Eligibility Criteria and settings, Interventions, Outcomes, sample size/power considerations, Interim analysis and stopping rules, Randomization (details on random number generation, allocation concealment, implementation), Blinding issues, etc, should be mentioned. The authors are advised to create separate subsections for each of the possible topics (whichever necessary), and that way produce a very clear writeup. I see the Authors indeed made an attempt; however, they are advised to write it carefully, following nice examples in the manuscript below:

https://www.sciencedirect.com/science/article/pii/S0889540619300010

Response: We reorganized the section based on the example provided. Please see highlighted titles in pages 5-9. 

Specific comments:

(a) For instance, the randomization and allocation concealment should be made very clear (they are NOT the same thing); the trial staff recruiting patients should NOT have the randomization list. Randomization should be prepared by the trial statistician, and he/she would not participate in the recruiting.

Response: Randomization was conducted within the computer program in real time. Because randomization and both intervention conditions were computer delivered, research staff stayed blind to intervention condition throughout the study. We have now clarified this section and also added a separate section on blinding as suggested. Please see page 8 lines 187-190 and page 9, lines 210-216.

(b) The manuscript does not provide details on the randomization procedure; it just states as a "standard" procedure. Provide more details. Why was block randomization (BR) used? If not, any specific reason, why? This is because BR is often recommended in clinical studies/trials to ensure a balance in sample size across groups over time.

https://www.ncbi.nlm.nih.gov/pmc/articles/PMC2267325/

Response: The article did not mention block randomization. As noted above, we have added clarification and reorganized the section based on the example provided above. Please see the pages mentioned above. 

(c) Sample size/power: A sample size/power statement is still required even in a randomized pilot trial; see

https://www.sciencedirect.com/science/article/abs/pii/S0895435612002740

It's better if that is provided, utilizing the primary response, acceptable effect size, and at 90%, or 95%. Even pilot trials need to be conducted with some ballpark number. It should also be described in a separate sub-section.

Response: This has been added. Please see page 13, lines 294-305. 

(d) Statistical Analysis: Based on the (longitudinal) design of the study, the authors justifiably conducted a mixed linear model analysis. Who were the underlying Gaussian assumptions (of the random effect & error) assessed? Furthermore, a generalized estimating equation (GEE) would also be adequate; any reason why that was not conducted? (I am not asking authors to do it).

Response: The reviewer is correct. We have re-analyzed the data with models more appropriate to the very skewed and zero-inflated nature of most of the dependent variables in the study. The manuscript has been revised accordingly. Please see the description in the methods section page 13, lines 294-305 and the revisions reflected throughout the manuscript including the tables. 

2. Results & Conclusions:

(a) The authors should check that any statement of significance should be followed by a p-value in the entire Results section. Otherwise, the Results section look OK.

Response: we have added p-values as suggested. Please see tables 2 and 3. 

Reviewer #2: The study is addressing two important public health challenges in women's health, IPV and substance use. The use of technology-based interventions in the domestic violence shelter is justified given the multiple needs and overburdened staff. However, the manuscript would be strengthened with more details on the intervention (screen shots), details on the content related to substance use, given there may be use of multiple substances, etc. For example, what skills building, etc. are the focus for behavioral change? 

Response: unfortunately, we no longer have access to make screenshots of the intervention. However, we have described the intervention in this paper to explain its key components. Please see pages 6 and 7. 

Does the intervention provide the user with specific local referrals to service organizations in the community, what is the link between the intervention content and increase daily substance abuse treatment use? 

Response: the intervention did not provide referrals to local organizations. However, safety monitoring was conducted at each assessment session to identify and promptly refer those who might need additional services, regardless of their intervention status. Those who reported IPV or other safety concerns were referred to additional services. In the revised analysis, there is no longer a link between the computerized intervention and use of substance use treatment services. 

The intervention describes a personalized safety plan, does this mean that the safety plan includes strategies to seek treatment or discussing substance use with counselors for example? 

Response: the intervention offered participants the option to create a personalized Safety Plan that included specific goals and strategies for change that were part of the content of the intervention. We have added this detail in the intervention description. Please see page 7, lines 169-174. 

Are advocates/case managers in the shelter trained on the intervention, would be likely that a participant would discuss the information provided for behavior change and/or discuss/review personalized safety plan with an advocate, how was the advocate involvement in supporting participant taking into consideration when examining the implications of the findings? 

Response: The intervention did not involve shelter staff. Participants completed the intervention on their own in a private office at the shelter. 

More detail on the choice of control condition is needed, as it appears the control condition is not related to behavioral change but rather on time that equal intervention group. All the women are in shelter, it would be interested to understand what other services they are receiving during the stay that may also influence findings, such as case management, legal support, child care, employment assistance, substance abuse treatment referrals, etc. More discussion is needed related to increase in self-efficacy in control group.

Response: the control condition controlled for time spent on the computer-based intervention, maintain blinding of research assistants, and mimic the interactivity of the computer-based intervention condition[1]. Throughout the study, participants received other standard services they are eligible for regardless of their intervention condition. We have included this detail in the manuscript. Please see page 8 lines 187-192. 

 Further, more discussion is needed related to the findings on IPV, it appears that IPV perpetration over time has a pattern towards significance with intervention group reporting more perpetration than control group, and control condition appears to have greater reduction in IPV victimization, cyberbullying and perpetration than intervention, although not significant - it would be interested in more discussion about these findings given IPV severity is stated outcome. 

Response: we did not propose IPV outcomes as main outcomes because this was a feasibility and acceptability study with a small sample size. Results can be swayed a lot by one or two people due to the study being under powered. The intervention focused on their drug use, as they were already receiving services targeting their IPV. 

Further, would be useful to report that participants in control condition used more safety strategies than intervention, this may not matter as use does not indicate that strategy was helpful/useful, so think need more discussion of these findings and how they inform future versions of the intervention. For example, the lower use of strategies by intervention group may be related to the personalized strategies so intervention participants did not need to use more strategies.

Response: these findings are exploratory, from a small sample size, hence, may be difficult to explain them further. We have added a paragraph in the discussion section clarifying this. Please see page 21, lines 420-426.

I do think detailing a framework for feasibility/acceptability testing would be useful, what was the measure used for acceptability for the intervention? I would recommend a table of the feasibility/acceptability, outcome and mediator measures used for study, would make it easier to read and follow. 

Response: results related to feasibility and acceptability of the computerized intervention are presented in table 2 and pages 16-17. 

More information on human subjects protections given that many shelters do not allow women to use substance, what strategies were used if a woman was active user?

Response: This is seldom the case anymore. Use of substances does not lead to discharge from most shelters currently. We have submitted the IRB protocol with details on human subjects protections as a supplemental file. 

The study did not detail limitations, small non generalizable sample from 3 different shelters, no race/ethnic difference assessed related to substance use, IPV severity, etc. No discussion of reasons why control group had significantly improved self-efficacy? 

Response: we have expanded the limitations section. The goal of this study was to establish feasibility and acceptability of the computerized intervention. Therefore, it was not powered for effectiveness outcomes, and effectiveness results should be interpreted with caution. We have included this statement in the limitations of the paper. Furthermore, the self-efficacy finding was no longer statistically significant on re-analysis. Please see page 21, lines 420-426. 

Although the intervention was reviewed as feasible, there was little indication that the intervention would be effective given the outcomes of reduce substance use, IPV severity and self-efficacy, what additional work, like qualitative interviews, etc will need to be done to revise the intervention to meet the needs of survivors?

Response: Evaluating the effectiveness was not the goal of this study; feasibility and acceptability was the goal of this study. This study is underpowered to show changes in those outcomes. A future effectiveness study with a large sample size is needed. 

1. Wernette GT, Plegue M, Kahler CW, Sen A, Zlotnick C: A pilot randomized controlled trial of a computer-delivered brief intervention for substance use and risky sex during pregnancy. Journal of Women's Health 2018, 27(1):83-92.

Best wishes,

Maji Hailemariam, PhD

---

## [Decision Letter · Decision Letter 1]

26 Apr 2023

Computer-based intervention for residents of domestic violence shelters with substance use: A randomized pilot study

PONE-D-22-28122R1

Dear Dr. Hailemariam,

We’re pleased to inform you that your manuscript has been judged scientifically suitable for publication and will be formally accepted for publication once it meets all outstanding technical requirements.

Kind regards,

Sandra Ann Springer, M.D.

Academic Editor

PLOS ONE

Additional Editor Comments (optional):

The authors have carefully responded to the reviewers' concerns and questions and the article is not acceptable for publication.

Reviewers' comments:

Reviewer's Responses to Questions

**Comments to the Author**

1. If the authors have adequately addressed your comments raised in a previous round of review and you feel that this manuscript is now acceptable for publication, you may indicate that here to bypass the “Comments to the Author” section, enter your conflict of interest statement in the “Confidential to Editor” section, and submit your "Accept" recommendation.

Reviewer #1: All comments have been addressed

2. Is the manuscript technically sound, and do the data support the conclusions?

Reviewer #1: (No Response)

3. Has the statistical analysis been performed appropriately and rigorously? 

Reviewer #1: (No Response)

4. Have the authors made all data underlying the findings in their manuscript fully available?

Reviewer #1: (No Response)

5. Is the manuscript presented in an intelligible fashion and written in standard English?

Reviewer #1: (No Response)

6. Review Comments to the Author

Reviewer #1: (No Response)

7. PLOS authors have the option to publish the peer review history of their article (what does this mean?). If published, this will include your full peer review and any attached files.

Reviewer #1: No

---

## [Editor Report · Acceptance letter]

17 May 2023

PONE-D-22-28122R1 

Computer-based intervention for residents of domestic violence shelters with substance use: A randomized pilot study 

Dear Dr. Hailemariam:

I'm pleased to inform you that your manuscript has been deemed suitable for publication in PLOS ONE. Congratulations! Your manuscript is now with our production department. 

Kind regards, 

on behalf of

Dr. Sandra Ann Springer 

Academic Editor

PLOS ONE